# Increase of MAL-II Binding Alpha2,3-Sialylated Glycan Is Associated with 5-FU Resistance and Short Survival of Cholangiocarcinoma Patients

**DOI:** 10.3390/medicina55120761

**Published:** 2019-11-28

**Authors:** Sasiprapa Wattanavises, Atit Silsirivanit, Kanlayanee Sawanyawisuth, Ubon Cha’on, Sakda Waraasawapati, Waraporn Saentaweesuk, Sukanya Luang, Chalongchai Chalermwat, Chaisiri Wongkham, Sopit Wongkham

**Affiliations:** 1Department of Biochemistry, Faculty of Medicine, Khon Kaen University, Khon Kaen 40002, Thailand; w_sasiprapa@kkumail.com (S.W.); kanlayanee@kkumail.com (K.S.); ubocha@kku.ac.th (U.C.); sukany@kku.ac.th (S.L.); cchanl@kku.ac.th (C.C.); chaisiri@kku.ac.th (C.W.); sopit@kku.ac.th (S.W.); 2Cholangiocarcinoma Research Institute, Khon Kaen University, Khon Kaen 40002, Thailand; sakdawa@kku.ac.th; 3Department of Pathology, Faculty of Medicine, Khon Kaen University, Khon Kaen 40002, Thailand; 4Faculty of Pharmacy, Mahasarakham University, Maha Sarakham 44150, Thailand; waraporn.sa@msu.ac.th

**Keywords:** cancer, chemotherapy, glycosylation, lectin, sialylation

## Abstract

*Background and objectives*: Sialylation plays important roles in tumor progression. Our present study aimed to demonstrate the alteration of sialylation and its role in cholangiocarcinoma (CCA). *Materials and Methods*: The α2,3- and α2,6-sialylation in CCA tissue was analyzed by lectin-histochemistry using *Maackia amurensis* lectin-II (MAL-II) and Sambucus nigra agglutinin (SNA). CCA cell lines were treated with the pan-sialylation inhibitor 3Fax-peracetyl-Neu5Ac (3F-Sia) followed by proliferation and chemosensitivity assays. *Results*: MAL-II binding α2,3-Sialylated Glycan (MAL-SG) and SNA binding α2,6-Sialylated Glycan (SNA-SG) were both elevated in CCA compared with hyperplastic/dysplastic (HP/DP) and normal bile ducts (NBD). The positive staining for MAL-SG or SNA-SG were found in 82% (61/74) of the CCA cases. Higher expression of MAL-SG in CCA was associated with shorter survival of the patients. The median survival of patients with high and low MAL-SG were 167 and 308 days, respectively, with overall survival of 233 days, suggesting the involvement of MAL-SG in CCA progression. MAL-SG expression of CCA cell lines was markedly decreased after treatment with 3F-Sia for 48 to 72 h. While proliferation of CCA cells were not affected by 3F-Sia treatment, their susceptibility to 5-fluorouracil (5-FU) was significantly enhanced. These results suggest that sialylation is involved in the development of 5-FU resistance and the sialylation inhibitor 3F-Sia can be used as a chemosensitizer for CCA. *Conclusions*: Sialylation is critically involved in the development of chemoresistance of CCA, and sialylation inhibitors may be used as a chemosensitizer in CCA treatment.

## 1. Introduction

Cholangiocarcinoma (CCA), a malignant tumor originated from bile duct epithelia, is highly endemic in the Northeastern Thailand and also arising worldwide [1]. Because clinical symptoms of CCA are not specific, most of the patients are detected at the advanced stage where metastasis and multi-drug resistance has already been developed. Many recent studies demonstrate the association between aberrant glycosylation and CCA progression [2,3,4,5].

Sialylation is a process of adding a terminal sialic acid (Sia) to the subterminal monosaccharide of carbohydrate chains on glycoproteins or glycolipids. It plays many roles in normal physiology, for example, neural differentiation [6], tissue regeneration [7,8], and resistance to influenza virus infection [9,10]. The increase of sialylation and sialylated-glycans were associated with development and progression of human diseases including cancer [10,11,12,13,14,15]. Sialylation plays important roles in tumor metastasis of many types of cancer such as colon [16], thyroid [14], and melanoma [17]. In addition, sialylation is involved in chemoresistance of ovarian [18,19], gastric [20], and colon [21] cancers.

Sialic binding lectins, such as *Maackia amurensis* lectin-II (MAL-II, α2,3-sialylated glycan binding lectin) and *Sambucus nigra* agglutinin (SNA, α2,6-sialylated glycan binding lectin), have been used for detecting and studying biological roles of sialylated-glycans in human diseases [9,11,12,14,15,17,22,23,24,25].

In this study, we have determined the expression of MAL-II binding α2,3-Sialylated Glycan (MAL-SG) and SNA binding α2,6-Sialylated Glycan (SNA-SG) in CCA tissue using lectin histochemistry. Correlation of MAL-SG and SNA-SG levels with clinical parameters and survival of the patients was evaluated. In addition, roles of sialylation in CCA progressions were determined using CCA cell lines.

## 2. Materials and Methods

### 2.1. CCA Tissues from Patients

Paraffin-embedded CCA tissue (*n* = 74) were obtained from the specimen bank of the Cholangiocarcinoma Research Institute, Khon Kaen University, Thailand. Informed consent was obtained individually from each subject and the experimental protocols were approved by the Human Ethics Committee of Khon Kaen University (HE571283 and HE591308).

### 2.2. Cholangiocyte and CCA Cell Lines

MMNK, an immortalized cholangiocyte cell line [26], was obtained from the Japanese Collection of Research Bioresources Cell Bank (JCRB) through the Cholangiocarcinoma Research Institute, Khon Kaen University, Thailand. CCA cell lines, KKU-213 and KKU-214, were established from a primary tumor of CCA patients and deposited in JCRB. KKU-213L5 and KKU-214L5 were the lung metastatic CCA cell lines derived from KKU-213 and KKU-214 as previously described [27,28]. All cell lines were cultured in Dulbecco’s Modified Eagle Medium (DMEM) supplemented with 10% heat-inactivated fetal bovine serum (FBS) and antibiotic-antimycotic in a 5% CO_2_ incubator at 37 °C.

### 2.3. Lectin-Histochemistry Staining

Lectin-histochemistry staining to detect MAL-SG and SNA-SG in CCA tissue was processed as previously described [3]. In brief, CCA tissue sections were de-paraffinized, re-hydrated, and incubated with 40 µg/mL of biotinylated-MAL-II and 1 µg/mL biotinylated-SNA (Vector Laboratories, Burlingame, CA, USA), respectively. Negative control slides were incubated with phosphate buffer saline (PBS) instead of biotinylated-lectin. Expression of MAL-SG and SNA-SG in CCA tissues was semi-quantified as a MAL-SG score and a SNA-SG score, according to their staining intensity (0, negatively stained; 1+, weakly stained; 2+, moderately stained; and 3+, strongly stained) and frequency of each intensity (% of total area) based on the H-Score system [29].

### 2.4. Lectin-Cyto-Fluorescence Staining

Lectin-cyto-fluorescence staining was used to detect MAL-SG in cultured cell lines. After treatment with a sialyltransferase inhibitor, cells were washed twice with ice-cold PBS and fixed with methanol for 30 min. PBS containing 3% bovine serum albumin (BSA) was used as a blocking buffer. Cells were incubated overnight at 4 °C with 80 µg/mL of biotinylated-MAL-II (Vector Laboratories, Burlingame, CA, USA) followed by 40 min incubation with 1:500 Alexa488-conjugated streptavidin (Invitrogen, Carlsbad, CA, USA) in PBS at room temperature. Nucleus was counter-stained with 1:10,000 diluted Hoechst33342 (Invitrogen, Carlsbad, CA, USA) and the signal was observed under a ZEISS LSM 800 Confocal Laser Scanning Microscope (Zeiss, Oberkochen, Germany).

### 2.5. Cell Proliferation and Chemosensitivity Assay

Roles of sialylation in cell proliferation and chemosensitivity were investigated using CCA cell lines. Cells were seeded in a 96-well culture plate, cultured overnight, and then treated with 50 µM of the pan-sialyltransferase inhibitor 3Fax-peracetyl-Neu5Ac (3F-Sia, Merck Millipore, Billerica, MA, USA) for 48–72 h. To determine the effects of 3F-Sia on CCA cell proliferation, cell number was measured at 0 h and 72 h after 3F-Sia treatment using Cell Counting Kit-8 (CCK-8, Dojindo Laboratories, Kumamoto, Japan) according to the manufacturer’s recommendation. To determine the effect of 3F-Sia on chemosensitivity to 5-fluorouracil (5-FU; Sigma Aldrich, Irvine, UK) of CCA cell lines, cells were treated with 50 µM 3F-Sia for 48 h, and then treated with 10 µM of 5-FU for an additional 48 h. Cell viability was measured at 0 and 48 h after 5-FU treatment. Cells treated with dimethyl sulfoxide (DMSO), instead of 3F-Sia, were used as a control. Experiments were performed in 5 replicates and repeated at least twice; the data presented in this study were from a representative experiment.

### 2.6. Statistical Analysis

Statistical analysis was performed using GraphPad Prism^®^ 8.0 (GraphPad software, Inc., La Jolla, CA, USA) and SPSS 17.0 (SPSS, Chicago, IL, USA). A Student’s *t*-test was used to evaluate the expression of MAL-SG and SNA-SG in CCA tissue, and the effect of 3F-Sia on CCA cell proliferation and chemosensitivity. The correlation of MAL-SG and SNA-SG expression and clinical parameters of CCA patients were analyzed using a χ^2^ (chi-square) test. Survival analysis was performed using Log-rank test and a Kaplan-Meier plot. Significant differences were considered by *p* < 0.05.

## 3. Results

### 3.1. MAL-SG and SNA-SG Were Elevated in CCA Compared with Normal Bile Ducts and HP/DP

Expression of MAL-SG and SNA-SG in 74 histologically proven CCA tissues were examined. MAL-SG was undetectable in hepatocytes and normal bile ducts (NBD) in the normal tissues adjacent to CCA. It was slightly expressed in hyperplastic/dysplastic bile ducts (HP/DP, median MAL-SG score = 0) and highly expressed in CCA (median MAL-SG score = 50; *p* < 0.05, Student’s *t*-test; Figure 1a,b). The staining intensity of MAL-SG in CCA varied from negative to strongly positive (3+) as shown in Figure 1a. The positive staining of MAL-SG was found in 82% (61/74) of CCA patients, with 38% (28/74) having s high MAL-SG score (101–300), 34% (25/74) a moderate MAL-SG score (11–100), and 28% (21/74) a negative to low MAL-SG score (0–10). SNA-SG was weakly expressed in NBD (median SNA-SG score = 5) and was moderately expressed in HP/DP (median SNA-SG score = 10) and CCA (median SNA-SG score = 20; *p* < 0.05, Student’s *t*-test; Figure 1a,c). Positive signal of SNA-SG was found in 82% (61/74) of CCA. Among them, 14% (10/74) had a high SNA-SG score (101–300), 45% (33/74) a moderate SNA-SG score (11–100), and 42% (31/74) a negative to low SNA-SG score (0–10).

### 3.2. High Level of MAL-SG in CCA Was Associated with Shorter Survival of CCA Patients

Correlation of MAL-SG and SNA-SG expression in CCA tissue with the clinical parameters of CCA patients was analyzed using a χ2 (chi-square) test. The patients were divided into high and low expression groups based on the median MAL-SG or SNA-SG scores in CCA tissues. Our data showed that expressions of MAL-SG or SNA-SG was not correlated with any age, histological types, or tumor stages of CCA patients (Table 1). High SNA-SG levels were more frequently observed in females than in males (*p* = 0.022).

Kaplan-Meier plots and Log-rank test were used to analyze the correlation of MAL-SG and SNA-SG levels with the survival of CCA patients. The data showed that survival of patients with high MAL-SG (MAL-SG score ≥ 50) was shorter than those with low MAL-SG (MAL-SG score < 50) (*p* < 0.05, Figure 1d). The median survival of patients with high MAL-SG was 167 days (95% CI, 94–239 days), whereas that of patients with low MAL-SG was 308 days (95% CI, 252–363 days). Different from MAL-SG, the SNA-SG expression level was not correlated with the survival of CCA patients (Figure 1e), as median survivals of patients with high SNA-SG (SNA-SG score ≥ 20) and low SNA-SG score (<20) were 233 days (95% CI, 158–307 days) and 236 days (95% CI, 90–381 days), respectively. Overall survival of CCA patients was 233 days with 95% CI of 165–300 days. Multivariate survival analysis using the Cox-proportional hazard model revealed that a high level of MAL-SG independently predicts the shorter survival of CCA patients regardless of age, sex, histological types, and tumor stages (*p* < 0.05). The hazard ratio of patients with high MAL-SG was 1.9 times higher than those with low MAL-SG (Table 2).

### 3.3. Suppression of Sialylation by a Sialyltransferase Inhibitor Altered the Expression of MAL-SG

As only the expression of MAL-SG was associated with poor clinical outcome of CCA patients, further experiments were focused on MAL-SG only. Expression of MAL-SG in normal cholangiocyte (MMNK1) and CCA cell lines (KKU-055, KKU-213, KKU-213L5, KKU-214, KKU-214L5) was determined using MAL-II lectin-cyto-fluorescence. The expression of MAL-SG varied among MMNK1 and CCA cell lines. MMNK1 and KKU-055 expressed a low level of MAL-SG, whereas KKU-213, KKU-213L5, KKU-214, and KKU-214L5 exhibited a high expression of MAL-SG (Figure 2). To see the roles of sialylation in CCA cell proliferation, 50 µM of 3F-Sia sialyltransferase inhibitor was used to inhibit the sialylation of high MAL-SG expressed KKU213, KKU213-L5, KKU214, and KKU214-L5 cell lines. The expression of MAL-SG in CCA cell lines was dramatically decreased after treatment with 3F-Sia for 48 h, and the suppressive effect persisted until 72 h (Figure 3a). In contrast, proliferation of 3F-Sia-treated CCA cell lines was comparable with that of DMSO-treated control cells (Figure 3b).

### 3.4. Suppression of Sialylation Enhances the 5-FU Susceptibility of CCA Cell Lines

To elucidate the role of sialylation on 5-FU susceptibility of CCA cell lines, they were treated with 50 µM 3F-Sia for 48 h. By this treatment, MAL-SG expression obviously decreased compared with 1% DMSO-treated control cells. Then, the cells were treated with 10 µM 5-FU and cell viability was measured 48 h later. The results showed that, after 5-FU treatment, the viability of 3F-Sia-treated (sialylation suppressed) cells was significantly lower than those of DMSO-treated (high sialylation) controls (*p* < 0.05, Student’s *t*-test; Figure 3c).

## 4. Discussion

Aberrant sialylation, either α2,3- or α2,6-sialylation, has been reported in various cancers, such as oral [30], ovary [31], prostate [32], and gastric [33] cancer. In CCA, sialyl Lewis-A (sLe^a^) or CA19-9 and total sialic acids were elevated in the patients’ sera, and it is currently used as a tumor marker for detection of CCA [34,35]. Cell surface sLe^a^ was found to play important roles in CCA metastasis, because the neutralization of sLe^a^ by a specific antibody could suppress in vitro metastatic ability of CCA cells [2]. These data suggested the possible increase of sialylation in CCA, although the direct evidence has never been documented. In this study, α2,3 and α2,6-sialylation status of NBD, HP/DP and CCA were investigated using lectin-histochemistry. The results showed that both MAL-SG (α2,3-sialylated glycan) and SNA-SG (α2,6-sialylated glycan) expression were higher in CCA compared to NBD and HP/DP, suggesting the possible association between the increase of sialylation and CCA development. Although the mechanisms underlying the increase of sialylation in CCA were not yet clearly defined, it is possible to be triggered by the increase of sialic acid synthesis or sialyltransferases and/or the decrease of sialidases, as was shown previously in other cancers [33,36,37].

As was previously shown, the serum CA19-9 and total sialic acids levels were increased in CCA patients compared with healthy persons, as such serum sialic acids have been used as the biomarkers for the diagnosis of CCA [34,35]. In this study, MAL-SG and SNA-SG was highly detected in CCA whereas they were very low in NBD, this information suggests the possibility of MAL-SG and SNA-SG as tumor markers for CCA. Thus, further experiments are necessary to investigate the possibility of detecting MAL-SG or SNA-SG in serum samples, as well as their diagnostic values for CCA. Identification of carrier proteins for MAL-SG and SNA-SG in patients’ tissue and sera is also important for understanding the biology of MAL-SG and SNA-SG in CCA. Development of lectin-based ELISA to measure MAL-SG and SNA-SG in clinical samples is necessary to evaluate their diagnostic values for CCA.

An association between the increased sialylation and cancer progression has been demonstrated in various cancers as it plays important roles in metastasis, immune evasion, radioresistance, and chemoresistance [21,37,38,39]. Elevation of α2,3-sialylated-glycans via increase of α2,3-sialyltransferase was associated with metastasis of gastric cancer [33] and the advanced stage of ovarian cancer [31]. Progression of prostate cancer was promoted by the increase of α2,6-sialylated-glycans via the overexpression of ST6GAL1 [40]. In this study, higher expression of MAL-SG in CCA was associated with shorter survival of the patients. To elucidate the roles of sialylation in CCA progression, we have suppressed sialylation of CCA cell lines by 3F-Sia, a pan-sialyltransferase inhibitor [41]. 5-FU has been used clinically as a backbone for treatment of CCA as it is an inexpensive chemotherapeutic drug. The use of 5-FU in combination with other agents is known to improve its effectiveness on CCA treatment, although the responsive rate varies between 7–43% [42]. Therefore, in this study we investigated the possibility of using 5-FU in combination with 3F-Sia to improve the CCA treatment. The results showed that suppression of sialylation by 3F-Sia augmented the chemosensitivity of CCA cell lines to 5-FU. These results suggest for the first time a role of sialylation in 5-FU resistance of CCA. The role of sialylation in chemoresistance has already been reported in many cancers [18,19,21]. Not only against 5-FU, sialylation was involved also in the resistance to paclitaxel and cisplatin of ovarian cancer cells [18,43]. Moreover, gefitinib resistance of ovarian cancer [18] and colon cancer [21] was promoted by sialylation. Taking all these and our present data together, sialylation inhibitor(s) can be used as a chemosensitizer in the new therapeutic strategy for CCA treatment. The mechanisms by which sialylation regulates chemoresistance of cancer cells are not well documented. It was reported that glycosylation is important for the function of membrane drug transporters [44,45,46]. It is speculated that aberrant glycosylation of ABC transporters may increase their activity and contribute to the resistance to chemotherapeutic drugs of cancer cells [44,46]. Taking this information together with our present data, the possible mechanisms of sialylation in 5-FU resistance of CCA cells can be explained as such that hypersialylation of ABC transporters increases their activity to export the chemotherapeutic drugs out of the cells. Further in vitro and in vivo studies are needed to address this hypothesis.

## 5. Conclusions

We have demonstrated here the increase of α2,3- and α2,6-sialylation in CCA using lectin-histochemistry for MAL-SG and SNA-SG. The higher expression of MAL-SG was associated with shorter survival of the patients, suggesting that MAL-SG can be a candidate of a poor prognostic marker for CCA. Functional analysis suggested that sialylation is involved in chemosensitivity of CCA, because sialylation inhibition could increase chemosensitivity of CCA to 5-FU. The present results provide the evidence of the increase of sialylation in CCA, which might be a target for improvement of CCA chemotherapy to bring better quality of life for CCA patients.

## Figures and Tables

**Figure 1 medicina-55-00761-f001:**
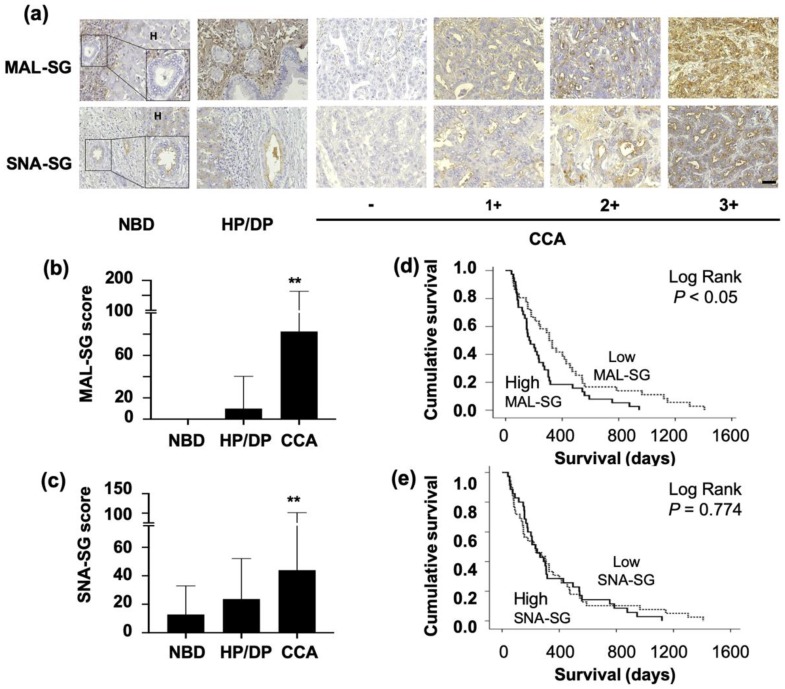
Expression of MAL-II binding α2,3-Sialylated Glycan (MAL-SG) and SNA binding α2,6-Sialylated Glycan (SNA-SG) in CCA tissues. (**a**) Lectin-histochemistry staining by *Maackia amurensis* lectin-II (MAL-II) and *Sambucus nigra* agglutinin (SNA) were performed in 74 histological-proven CCA tissues. (**b**,**c**) Expression of MAL-SG and SNA-SG were presented as lectinhistochemistry (LHC) score base on the staining frequency and intensity. (**d**,**e**) Survival analysis of CCA patients was performed using Kaplan-Meier plots and Log-rank tests according to MAL-SG and SNA-SG scores in tumor area. **significant difference, *p* < 0.001.

**Figure 2 medicina-55-00761-f002:**
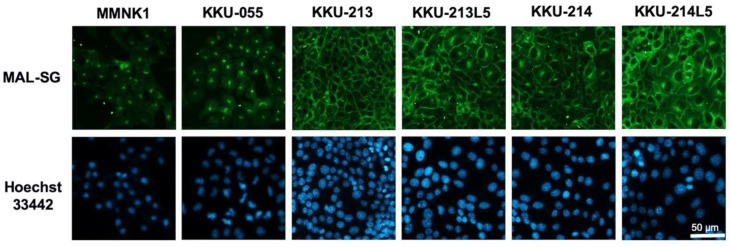
Expression of MAL-SG in CCA cell lines. MAL-SG expression in MMNK1 and CCA cell lines (KKU-213, KKU-214, KKU213-L5, and KKU214-L5) was determined by MAL-II lectin-cyto-fluorescent staining. The signal of Alexa-448 represented MAL-SG (green) and nucleus was stained by Hoechst-33342 (blue).

**Figure 3 medicina-55-00761-f003:**
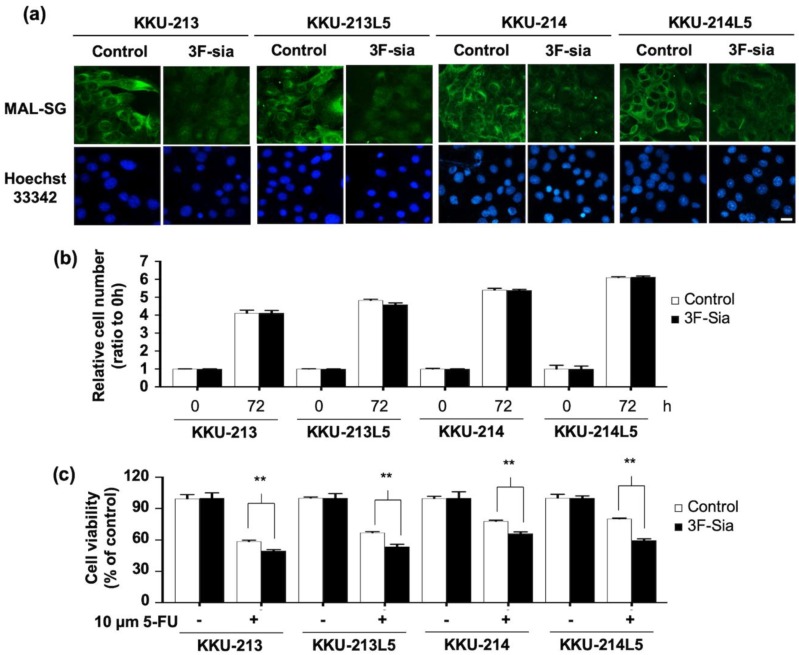
Effect of 3Fax-peracetyl-Neu5Ac (3F-Sia) on MAL-SG expression. 3F-Sia, a sialyltransferase inhibitor, was used to suppress the expression of MAL-SG in CCA cell lines. After 48 h of 50 µM 3F-Sia treatment, (**a**) MAL-SG was determined by lectin-cyto-fluorescent staining, the signal of MAL-SG was shown by Alexa-448 (green), and nucleus was stained by Hoechst-33342 (blue). (**b**) Proliferation of CCA cell lines was measured at 0 and 72 h after 3F-Sia treatment by Cell Counting Kit-8. After 3F-Sia treatment, the cells were treated with 10 µM of 5-FU for another 48 h. (**c**) Cytotoxicity was measured at 0 and 48 h after 5-FU treatment by WST assay.

**Table 1 medicina-55-00761-t001:** Correlation of MAL-SG and SNA-SG expression and clinical data of CCA patients.

Variables	*n*	MAL-SG			SNA-SG		
		Low	High	*p*	Low	High	*p*
		(<50)	(≥50)		(<20)	(≥20)	
Histological type (*n* = 74)				0.242			0.474
Papillary	22	13	9		13	9	
Non-papillary	52	23	29		26	26	
Age (years) (*n* = 74)				0.496			0.668
≤56	34	18	16		17	17	
>56	40	18	22		22	18	
Gender (*n* = 74)				0.864			0.022
Female	26	13	13		9	17	
Male	48	23	25		30	18	
Tumor size (*n* = 73)				0.887			0.233
<5 cm	13	6	7		5	8	
≥5 cm	60	29	31		34	26	
Tumor stage (*n* = 74)				0.814			0.963
I-III	29	15	14		15	14	
IVA	35	17	18		19	16	
IVB	10	4	6		5	5	

**Table 2 medicina-55-00761-t002:** Cox-proportional hazard model for multivariate survival analysis of MAL-SG in CCA patients.

Variables	*n*	Hazard Ratio (HR)	95% (CI)	*p*
Histological type (*n*=74)			1.117–3.572	0.020
Papillary	22	1
Non-papillary	52	1.997
Age (years) (*n* = 74)			0.844–2.288	0.195
≤56	34	1
>56	40	1.390
Gender (*n* = 74)			0.691–1.873	0.613
Female	26	1
Male	48	1.137
Tumor stage (*n* = 74)				
I-III	29	1	0.092
IVA	35	0.840	0.499–1.415	0.513
IVB	10	2.005	0.930–4.322	0.076
MAL-II expression (*n* = 74)			1.139–3.246	0.014
Low	36	1		
High	38	1.923

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
