# Peer review of "Increase of MAL-II Binding Alpha2,3-Sialylated Glycan Is Associated with 5-FU Resistance and Short Survival of Cholangiocarcinoma Patients"

_medicina, 2019, doi:10.3390/medicina55120761_

Round 1

Reviewer 1 Report

Notes:

line 48 gastric

line 90 4C

line 211, 223  Ref 30 is not appropriate.  Not stomach cancer and not sialyltransferase, respectively  are presented in that article. Please, correct it.

line 239 Ref 39 is not appropriate, since this paper is about α2,6-sialyltransferase and not α2,3-sialyltransferase as has been referred here.

line 241 Please insert reference.

Questions:

1.  Is there any difference between MAL-II bindings of KKU-213 and KKU-213L . According to Fig2 no difference  exists, but in Fig3a the control KKU-213L bind much more MAL-II. Could you please explain it?

2. What is the reason of spot-like staining with MALII  in MMNK1 and KKU-055 cells in Fig2?

Author Response

Reviewer 1

Q1: line 48 gastric

RE1: corrected as suggested

Q2: line 90 4C

RE: corrected to 4ËšC

Q3: line 211, 223 Ref 30 is not appropriate.  Not stomach cancer and not sialyltransferase, respectively are presented in that article. Please, correct it.

RE: We have added the appropriate reference (Ref. 31) instead of Ref. 30 to support the increase of alpha-2,3 sialylation in gastric (stomach) cancer as suggested by the reviewer. 

Q4: line 239 Ref 39 is not appropriate, since this paper is about α2,6-sialyltransferase and not α2,3-sialyltransferase as has been referred here.

RE: We have added the appropriate reference (Ref. 31) to support the association of alpha-2,3 sialylation and metastasis of gastric cancer.

line 241 Please insert reference.

RE: Ref. has been added as Ref. 42

Questions:

Is there any difference between MAL-II bindings of KKU-213 and KKU-213L5. According to Fig2 no difference exists, but in Fig3a the control KKU-213L5 bind much more MAL-II. Could you please explain it?

RE: In Fig. 2, we have just classified MMNK1 and KKU-055 as MAL-SG low expression cell lines and KKU-213, KKU-213L5, KKU-214, and KKU-214L5 as MAL-SG high expression cell lines. Again, in Fig. 3a, MAL-II staining of control KKU-213 and KKU-213L5 cells did not show any obvious difference. Similarly, no obvious difference of MAL-II staining was seen between KKU-214 and KKU 214L5 cell lines. In all four cell lines, MAL-II stainability was drastically suppressed by these two cell lines3F-Sia treatment. 

What is the reason of spot-like staining with MALII in MMNK1 and KKU-055 cells in Fig2?

RE: For staining with MAL-II, we have used the permeabilized (by MeOH) protocol, which facilitate the antibody or lectin to reach every part of cells including intracellular organelles.  The sialylation process is localized in Golgi, and, therefore, the use of the permeabilized-method for MALII staining possibly caused the positive staining in Golgi-area (peri-nuclear staining) as shown in MMNK1 and KKU-055 cells.  Expression of MAL-SG in MMNK1 and KKU-055 is relatively low, compare with other cell types, this may cause the spot-like staining in Golgi-area of the cells.  Other cells, such as KKU214 an dKKU-214L5) also show the positive signal in Golgi-area but the staining in membrane and cytoplasmic fractions were much stronger than spot-like staining of Golgi area.

Reviewer 2 Report

“Increase of MAL-II binding alpha2,3-sialylated glycan is associated with 5-FU resistance ahd short survival of cholangiocarcinoma patients” by Wattanavises et al. provides support for a role of aberrant glycosylation in cholangiocarcinoma patient outcome and perform some functional experiments to investigate the role of altered glycosylation with 5-FU chemosensitivity. The manuscript is very well-written and I only have a few suggestions that may help strengthen the authors’ conclusions and improve the clarity of the presentation.

Some discussion of the clinical use of 5-FU in treatment for CCA would be helpful in the introduction to provide context for the later studies. To determine if the effects of 3F-Sia on 5-FU chemosensitivity are truly due to on-target effects of the inhibitor, it would be useful to perform 5-FU +/- 3F-Sia treatment (as in Figure 3c) with cell lines that do not demonstrate MAL-11 reactivity (i.e., MMNK1 and KKU-055 cell lines) Some discussion of potential mechanism by which alternative glycosylation regulates chemosensitivity would be appropriate I noticed only a few small typos: Page 2, line 90: “4 oC” instead of “4oC”; Page 7, line 210: “Aberrant” instead of “A Aberrant”

Author Response

Reviewer 2

“Increase of MAL-II binding alpha2,3-sialylated glycan is associated with 5-FU resistance and short survival of cholangiocarcinoma patients” by Wattanavises et al. provides support for a role of aberrant glycosylation in cholangiocarcinoma patient outcome and perform some functional experiments to investigate the role of altered glycosylation with 5-FU chemosensitivity. The manuscript is very well-written and I only have a few suggestions that may help strengthen the authors’ conclusions and improve the clarity of the presentation.

1) Some discussion of the clinical use of 5-FU in treatment for CCA would be helpful in the introduction to provide context for the later studies. To determine if the effects of 3F-Sia on 5-FU chemosensitivity are truly due to on-target effects of the inhibitor, it would be useful to perform 5-FU +/- 3F-Sia treatment (as in Figure 3c) with cell lines that do not demonstrate MAL-11 reactivity (i.e., MMNK1 and KKU-055 cell lines).

RE: Thank you for the suggestion, the discussion on clinical use of 5-FU for CCA treatment has been added into Discussion, line 241 as

"5-FU has been used clinically as a backbone for treatment of CCA as it is an inexpensive chemotherapeutic drug.  The use of 5-FU in combination with other agents is known to improve its effectiveness on CCA treatment, although the responsive rate varied ranging 7-43% [42].  In this study, therefore, we investigated the possibility of using 5-FU in combination with 3F-Sia to improve the CCA treatment."

We thank the suggestion to add the data on the effects of 3F-Sia/5-FU in MAL-II low reactive cells (i.e., MMNK1 and KKU-055 cell lines).  Since it was previously reported that KKU-055 CCA cell lines is the most sensitive cell lines for many chemotherapeutic drugs including 5-FU, we did not investigate the effects of 3F-Sia on 5-FU sensitization on KKU-055 CCA cell line.  In addition, since MAL-SG expression of MMNK1 and KKU-055 were only weakly positive, we speculated that enhancement of the 5-FU responsiveness by 3F-Sia will be marginal in KKU-055 CCA cells or MMNK1 cells. In future, we will investigate the effect of 3F-Sia on 5-FU sensitivity of more CCA cell lines with high and low MAL-II expressing cells using dose response of 5-FU.

2) Some discussion of potential mechanism by which alternative glycosylation regulates chemosensitivity would be appropriate.

RE: Thank you for your suggestion. Accordingly, the discussion on glycosylation and chemoresistance has been added into Discussion, starting from line 253 as

" The mechanisms by which sialylation regulates chemoresistance of cancer cells are not well documented.   It was reported that glycosylation is important for the function of membrane drug transporters [44-46].  It is speculated that aberrant glycosylation of ABC transporters may increase their activity and contribute to the resistance to chemotherapeutic drugs of cancer cells [44,46].  Taken these information together with our present data, the possible mechanisms of sialylation in 5-FU resistance of CCA cells can be explained as such that hypersialylation of ABC transporters increases their activity to export the chemotherapeutic drugs out of the cells. Further in vitro and in vivo studies are still needed to address this hypothesis.

3) I noticed only a few small typos: Page 2, line 90: “4 oC” instead of “4oC”; Page 7, line 210: “Aberrant” instead of “A Aberrant” 

RE: Thank you very much, we have corrected those typos as suggested.
